# Transcutaneous Posterior Tibial Nerve Stimulation: An Adjuvant Treatment for Intractable Constipation in Children

**DOI:** 10.3390/biomedicines12010164

**Published:** 2024-01-12

**Authors:** Rebeca Mayara Padilha Rego, Nilton Carlos Machado, Mary de Assis Carvalho, Johann Souza Graffunder, Crhistiano Fraguas, Erika Veruska Paiva Ortolan, Pedro Luiz Toledo de Arruda Lourenção

**Affiliations:** 1Department of Surgery and Orthopedics, Division of Pediatric Surgery, Botucatu Medical School, São Paulo State University (UNESP), Botucatu 18618-687, SP, Brazil; mayara.padilha@unesp.br (R.M.P.R.); erika.ortolan@unesp.br (E.V.P.O.); 2Department of Pediatrics, Division of Pediatric Gastroenterology, Hepatology and Nutrition, Botucatu Medical School, São Paulo State University (UNESP), Botucatu 18618-687, SP, Brazil; nilton.machado@unesp.br (N.C.M.); mary.carvalho@unesp.br (M.d.A.C.); 3Botucatu Medical School, São Paulo State University (UNESP), Botucatu 18618-687, SP, Brazil; johann.graffunder@unesp.br; 4Dr. José Bahia Sapucaia Private Clinic, Salvador 41830-492, BA, Brazil; fraguasfisio@gmail.com

**Keywords:** constipation, transcutaneous electric nerve stimulation, child, adolescent

## Abstract

Background: Functional constipation can lead to painful defecations, fecal incontinence, and abdominal pain, significantly affecting a child’s quality of life. Treatment options include non-pharmacological and pharmacological approaches, but some cases are intractable and require alternative interventions like neuromodulation. A subtype of neuromodulation, called Transcutaneous Posterior Tibial Nerve Stimulation (TPTNS), comprises electrical stimulation at the ankle level, by means of electrodes fixed to the skin. TPTNS is a minimally invasive, easy-to-apply technique that can potentially improve constipation symptoms in the pediatric population by stimulating the sacral nerves. Aim: To evaluate the clinical results and applicability of TPTNS as an adjuvant treatment for children and adolescents with functional constipation. Methods: Between April 2019 and October 2021, 36 patients diagnosed with functional constipation according to the Rome IV Criteria were invited to participate in the study. The study followed a single-center, uncontrolled, prospective cohort design. Patients received TPTNS for 4 or 8 weeks, with assessments conducted immediately after the periods of TPTNS and 4 weeks after the end of the intervention period. The data normality distribution was determined by the Shapiro–Wilk test. The Wilcoxon test and Student’s *t*-test for paired samples were used to compare quantitative variables, and the McNemar test was used to compare categorical variables. Results: Of the 36 enrolled patients, 28 children and adolescents with intractable function constipation completed the study, receiving TPTNS for 4 weeks. Sixteen patients (57.1%) extended the intervention period for 4 extra weeks, receiving 8 weeks of intervention. TPTNS led to significant improvements in stool consistency, frequency of defecation, and bowel function scores, with a reduction in abdominal pain. Quality of life across physical and psychosocial domains showed substantial enhancements. The quality of life-related to bowel habits also improved significantly, particularly in lifestyle, behavior, and embarrassment domains. The positive effects of this intervention are seen relatively early, detected after 4 weeks of intervention, and even 4 weeks after the end of the intervention. TPTNS was well-tolerated, with an adherence rate of approximately 78%, and no adverse effects were reported. Conclusions: TPTNS is an adjuvant treatment for intractable functional constipation, improving bowel function and quality of life. The effects of TPTNS were observed relatively early and sustained even after treatment cessation.

## 1. Introduction

Constipation is a frequent cause of medical advice. About 3% of pediatric outpatient visits and approximately 25% of pediatric gastroenterology visits are related to defecation disorders, with a significant proportion of children having functional constipation and a minority (approximately 5%), having constipation associated with an identifiable organic cause [1,2]. In children with functional constipation, painful defecations, retentive fecal incontinence, and episodes of abdominal pain can impair their quality of life, affecting social and school life, with significant psychological consequences [3,4,5,6,7].

The standard treatment for children with functional constipation can be non-pharmacological or pharmacological [2,7,8,9]. In tertiary care centers, approximately 50% of children treated for functional constipation with the standard of care improved and can stop using laxatives within 6 to 12 months [10]. However, a portion of patients may not respond well to these measures. These children have severe and long-lasting symptoms, with recurrent fecal impaction, prolonged intervals between defecations, and large-volume stools, associated with episodes of fecal incontinence. When symptoms do not respond to conventional treatment for at least three months, the constipation is considered intractable [2,11]. Treatment options for these patients include injecting a botulinum toxin into the anal sphincter, retrograde or anterograde bowel lavage, neuromodulation, and colorectal resections [2,7].

Neuromodulation is a non-pharmacological approach that can improve intestinal motility by modulating colonic innervation [12]. A subtype of neuromodulation, called Transcutaneous Posterior Tibial Nerve Stimulation (TPTNS), comprises electrical stimulation at the ankle level, by means of electrodes fixed to the skin [13]. TPTNS is a minimally invasive, easy-to-apply technique that can potentially improve constipation symptoms in the pediatric population [13,14]. Electrostimulation of the tibial nerve can modulate urinary and defecation functions by stimulating the sacral nerves [13,14,15,16]. The applicability and effectiveness of TPTNS in the treatment of constipation and fecal incontinence in children have lately been studied [14,17]. Lecompte et al. [14] evaluated a group of 8 children with fecal incontinence associated with various underlying causes, including anorectal malformations, neurological diseases, and Hirschsprung’s disease. The authors found improved fecal incontinence after 6 months in 7 children, and resolution in 5 children [14]. One single study evaluated the effects of TPTNS in children with functional constipation. Velasco-Benitez et al. [17] showed significant improvement in fecal consistency and in episodes of retentive fecal incontinence, abdominal pain, and hematochezia in 20 patients with functional constipation who underwent daily TPTNS for 10 days.

For the above reasons, we decided to evaluate the clinical results and applicability of TPTNS as adjuvant treatment for children and adolescents with functional constipation.

## 2. Materials and Methods

### 2.1. Study Design and Scenario

We conducted a single-center, interventional, uncontrolled, prospective cohort study to evaluate the applicability and clinical outcomes of TPTNS in children and adolescents with functional constipation. This study was conducted at the Botucatu Medical School, São Paulo State University (UNESP), São Paulo, Brazil.

The patients underwent daily sessions of TPTNS for a period of 4 or 8 weeks and participated in semi-structured interviews, carried out at each of the assessment moments, in which aspects related to bowel habits, quality of life, and applicability of the intervention were evaluated. The study steps are summarized in Figure 1.

### 2.2. Ethical Aspects of the Study and Registry

This study was carried out in accordance with the principles determined by the Declaration of Helsinki, ISO14155, the Data Protection Act, and the Good Clinical Practice Guidelines. It was approved by the Internal Review Board (IRB) of the Botucatu Medical School, UNESP, São Paulo, Brazil, registration number CAAE 80013017.0.0000.5411. The patients and their guardians were previously informed about the purpose of the research and signed an informed consent form. Patients between 11 and 18 years old signed the respective informed assent form.

The study was registered in the Brazilian Registry of Clinical Trials (Rebec) under RBR-344jq8 and is available at http://www.ensaiosclinicos.gov.br (accessed on 2 January 2024). The study protocol, designed according to the recommendations of the Standard Protocol Items: Recommendations for Interventionist Tests—SPIRIT [18], was previously published [19].

### 2.3. Eligibility Criteria

The inclusion criteria were diagnosis of constipation according to the Rome IV criteria [20,21], age between 6 and 18 years, patient and/or guardian agreed and signed the informed consent form, and patients between 11 and 18 years agreed and signed the informed assent form. The exclusion criteria were presence of any organic cause of intestinal constipation, presence of neurologic and/or cognitive impairment, presence of skin lesions on the regions of the electrodes, abnormal sensitivity on the electrode’s region, parents or guardians not capable of understanding the training for the application of TPTNS, and children with heart diseases or arrythmias. Patients who did not complete the intervention period were also excluded.

### 2.4. Interventions

The patients underwent TPTNS based on the method described by Queralto et al. [22]. This intervention was supervised by a physiotherapist, experienced in this type of therapy, and a member of the research team. After local hygiene, a self-adhesive silicone electrode was applied approximately 3–4 cm above the medial malleolus of the tibia, and a second electrode to close the power circuit was placed just below the medial malleolus of the tibia on the same leg (Figure 2). This was repeated on the contralateral lower limb. The electrodes were then connected to an electrical stimulation device (De Tens/Fes-2 channels, Neurodyn Portable^®^, Ibramed, Amparo, SP, Brazil). The position of the electrodes was determined by noting the rhythmic flexion of the hallux before beginning the sessions. The intensity level of the current varied, being used at the intensity immediately below the patient’s maximum sensitivity threshold (usually between 10 and 30 mA). The current used was 200 μs and 20 Hz. The transcutaneous stimulation of the posterior tibial nerve was conducted simultaneously on both lower limbs, for 30 min daily, for 4 consecutive weeks, at the patient’s home. After each TPTNS session, the electrodes were removed. The self-adhesive silicone electrodes were changed every 10 sessions to minimize the potentially adverse effects of the therapy [22,23,24,25,26].

The parents or guardians were trained to perform TPTNS on their children. Training sessions were held, the application being supervised by the physiotherapist, until the parents felt confident enough to carry out the procedure without supervision. From that point on, TPTNS was carried out daily, at home. During the intervention period, the patients also participated in a supervised TPTNS session every 15 days, where the physiotherapist evaluated the technique and any doubts were cleared. In addition, the guardians had the telephone number of the research team to communicate any questions or events during the study period.

During the intervention period, patients maintained all therapeutic recommendations related to treating constipation (medications, diets, and behavioral counseling) previously prescribed by the medical team in charge. Patients were instructed to inform the research team about any changes in the treatment plan prescribed by the medical team during the intervention period.

### 2.5. Treatment Compliance and Adverse Events

Members of the research team investigated the possible occurrence of adverse events at all the assessment sessions and in the fortnightly sessions of supervised TPTNS. At any time during the study, patients/parents/guardians could contact the research team by telephone to communicate possible adverse events and clear any doubts. Patients with adverse events or clinical worsening of the defecation pattern, or who so requested, would be discontinued from the study.

### 2.6. Periods of Intervention and Assessment Moments

The intervention period lasted 4 or 8 consecutive weeks, initiated immediately after the end of the training period. After 4 weeks of TPTNS, the patients could decide if they wanted to have 4 more sessions, with 8 weeks of intervention. The moments chosen for the assessment interviews are as follows: Moment 0 (M0): 1 week before the beginning of the intervention; Moment 1 (M1): immediately after the period of 4 weeks of intervention; Moment 2 (M2): immediately after the period of 8 weeks of intervention (for patients who chose this duration of intervention); and Moment 3 (M3): 4 weeks after the end of the intervention period.

### 2.7. Assessment Interviews

At all assessment moments, patients and/or guardians participated in semi-structured interviews to determine the results of the intervention concerning bowel habits and quality of life. These interviews were conducted by the same member of the research team and lasted approximately 40 min. The following assessment instruments were applied: a questionnaire addressing the current clinical status (Appendix A) [19]; Modified Bristol Stool Form Scale for Children (mBSFS-C) to assess the stool consistency (Appendix A) [27,28,29]; Bowel Function Score (BF-S) to assess bowel function (Appendix A) [30]; Pediatric Quality of Life Inventory, version 4.0 (PedsQL 4.0) answered by caregivers, to assess the overall quality of life (Appendix A) [31,32]; and the questionnaire for the Assessment of Quality of Life in Children and Adolescents with Fecal Incontinence (AQLCAFI), answered by the children, to assess the quality of life in patients with intestinal symptoms (Appendix A) [33,34].

In addition, each patient was monitored using a standardized info diary form, with records regarding the frequency of defecation, stool consistency according to the mBSFS-C, frequency of episodes of fecal incontinence, of painful evacuations, of abdominal pain, presence of blood in the stools, use of laxatives and their doses, as well as possible adverse effects such as pain, nausea, vomiting, diarrhea, and flatulence. This diary was filled out for 7 days by the guardians, starting 7 days before each assessment visit. Patients using laxative medications had their doses monitored and any changes, both in increasing and in decreasing the dose, were recorded as possible outcomes for the study’s conclusions. During the assessment interview carried out at the end of the intervention, the guardians answered the questionnaire designed to assess the applicability of daily transcutaneous TPTNS at home (Appendix A) [19].

### 2.8. Analysis of Results

The analysis of the results was carried out, focusing on the assessment moments. This analysis used the clinical variables: number of defecations per week, fecal incontinence episodes per week, painful or hard bowel movements, and abdominal pain. The results obtained by the applied instruments (mBSFS-C, BF-S, PedsQL 4.0, and AQLCAFI) were also analyzed to assess the functional results of bowel habits and overall quality of life. All situations that might be indirectly involved with the clinical results of TPTNS, such as an increase or decrease in laxative doses or any clinical or dietary changes, and aspects related to the applicability and potential adverse effects of the intervention were also analyzed.

### 2.9. Sample Size Calculation

The estimated sample size was 28 patients, calculated for a paired *t*-test (before and after the intervention), based on an estimated mean increase of 7 defecations per month, according to data from previous TPTNS studies [15,35], an estimated standard deviation of change = 12, a 2-tailed alpha = 0.05, and a test power beta = 0.85.

### 2.10. Statistical Analysis

Statistical analysis was performed using SPSS 22.0 for Windows (SPSS Inc., Chicago, IL, USA). The characteristics of the patients included in the study were analyzed through descriptive statistical analysis. A comparative statistical analysis was carried out with the results obtained for the variables collected at the assessment moments. Continuous numerical data were expressed as mean ± standard deviation and median (first and third quartiles). The proportions were presented as percentages, with their respective confidence intervals. The comparison between the assessment moments was performed using different statistical tests, according to the data normality distribution, determined by the Shapiro–Wilk test. For statistical comparisons between quantitative variables, the Wilcoxon test and Student’s *t*-test for paired samples were used. The McNemar test was used to compare categorical variables. *p* values less than 0.05 were considered to indicate statistical significance. The statistical methods of this study were reviewed by Helio Rubens de Carvalho Nunes from Botucatu Medical School, São Paulo State University.

## 3. Results

Between April 2019 and October 2021, 36 patients diagnosed with functional constipation according to the Rome IV Criteria were invited to participate in the study [20,21]. A total of 28 patients finished the study, receiving TPTNS for 4 weeks. Sixteen patients (57.1%) extended the intervention period for 4 extra weeks, receiving 8 weeks of intervention. Figure 3 depicts the eligible children who finished the study.

### 3.1. Demographic and Clinical Characteristics

Table 1 depicts the sociodemographic characteristics of the patients who completed the study. The median age was 134 months (11 years), with an interquartile range of 105 to 145 months (8.7 to 12 years), characterizing a population of school children and adolescents. The patients’ families consisted of parents in their fourth decade of life, with a few children per couple, and who lived in small households. The mothers answered most of the questionnaires (79%).

Table 2 depicts the clinical characteristics of the patients who completed the study. The duration of symptoms was long, with a median of 8 years and an interquartile range of 3 to 10 years. All patients had refractory symptoms for more than 3 months, during which they had received conventional treatment for functional constipation, being classified as having intractable constipation according to the definition proposed by the European (ESPGHAN) and North American (NASPGHAN) Society for Pediatric Gastroenterology, Hepatology and Nutrition [2]. There was a high proportion of children with painful or hard bowel movements, and almost half of the patients often had abdominal pain.

In total, 23 (82.1%) patients were using laxatives regularly at baseline. The distribution of laxative use and any changes in dosages, introduction, or withdrawal of these medications during the study periods are listed in Table 3. During the study, 28 patients regularly used at least one of the following laxatives: sodium picosulfate, polyethylene glycol 3350, mineral oil, bisacodyl, or sennosides. After TPTNS, it was possible to decrease the dose or completely stop the medication in 7 (25%) patients. Four patients had an increase in the dose of laxative medications during the study. In all these cases, the increase occurred after the end of the intervention period and before the final assessment moment.

### 3.2. Analysis of TPTNS Effects on Bowel Function and Quality of Life

In total, 6 patients were excluded from the analyzes at Moment 3 (4 weeks after the end of TPTNS); 4 patients who had an increase in the dose of laxatives after the end of the intervention, and two patients who did not participate in the reassessment interview in M3.

There was a significant improvement in the indicators for stool consistency, frequency of defecation, and BF-S, as well as for the symptoms “hard bowel movement” and “painful bowel movements” when comparing moments M0 (pre-intervention) and M1 (immediately after 4 weeks of TPTNS) (Table 4). The significant improvement of these parameters was also seen when comparing M0 and M3 (4 weeks after the end of the intervention). There was a significant decrease in the distribution of “abdominal pain” between M0 and M1. On the other hand, no significant differences were identified for all parameters evaluated in the comparisons between M1 and M2 (after 8 weeks of TPTNS) and between M1 and M3.

Table 5 depicts the values obtained by applying the two instruments for assessing the quality of life in the four moments of the study, including the total score and the score for each of the respective domains. For both questionnaires, in all domains evaluated, there was a significant improvement in the indicators when comparing M0 (pre-TPTNS) and M3 (4 weeks after the end of the intervention). Most of the time, these differences could be identified as early as in the comparisons between M0 and M1 (immediately after 4 weeks of TPTNS). On the other hand, for most of the evaluated domains, no statistical differences were identified when comparing M1 and M2 (after 8 weeks of TPTNS) or between M1 and M3 (4 weeks after the end of the intervention).

### 3.3. Analysis of TPTNS Applicability

Of the 36 children who started the intervention proposed in the study, 28 children completed all the steps of TPTNS, and 8 children abandoned the study during the intervention phase, with an adherence rate of 77.8%. The caregivers of these 8 patients explained that they did not have time to supervise the intervention and therefore, stopped carrying it out. No patient had side effects or pain during the TPTNS.

Table 6 depicts the results obtained by applying the questionnaire to evaluate the applicability of TPTNS. Most parents thought the procedure was a positive experience, simple to conduct, with no difficulty, and that did not cause pain or discomfort to the children. The main difficulty pointed out by some parents was related to the time demanded for TPTNS every day. During the study, no adverse effects of the intervention were reported or identified.

## 4. Discussion

Our study consisted of school-age children and adolescents, with a median age of 11 years, most children of parents in their fourth decade of life, who lived in small households and received regular clinical care for the treatment of constipation at a referral center of the Brazilian public health system. These patients had chronic constipation, with a median duration of symptoms close to 8 years, had been undergoing standard treatment for constipation following the most recent guidelines, and were already regularly using at least one type of laxative [2,7,9]. Even with treatment, these patients were very symptomatic. Most presented defecation effort, anal pain to defecate, infrequent defecations, fecal retention, history of fecalomas, and elimination of bulky, hard stools. All patients could, therefore, be characterized as having intractable intestinal constipation, as they had persistent symptoms for more than three months, even after receiving conventional treatment [2].

In the last two decades, the experience with the use of neurostimulation in the treatment of children with gastrointestinal disorders has been growing, being considered a promising option for treating children with functional constipation refractory to conventional treatment [9]. Available neuromodulation strategies include techniques such as sacral stimulation, which requires the surgical implantation of devices or minimally invasive transcutaneous methods, such as interferential abdominal transcutaneous neurostimulation and sacral and posterior tibial transcutaneous electrostimulation, conducted using adhesives placed on the skin [13,36]. A recent systematic review with meta-analysis demonstrated that transcutaneous neuromodulation is an effective adjuvant treatment modality for children with constipation and retentive fecal incontinence. However, this review only included studies that used transcutaneous abdominal or sacral interferential electrical stimulation. No TPTNS intervention studies met the selection criteria and could be included in this meta-analysis [37].

TPTNS has a well-established role in the treatment of children with urinary dysfunction and adults with fecal incontinence [13,14,15,35,38,39,40]. There is evidence that TPTNS can stimulate the second and third sacral nerve roots, the same spinal cord segments that innervate the bladder, rectum, and pelvic floor, modulating bowel motility and aiding in treating constipation [14,36]. However, only one study specifically evaluated TPTNS as an adjunctive treatment for children with functional constipation. This study found significant clinical improvement. However, the intervention time was limited to only ten days, and no reassessments were conducted after the end of the intervention [17].

Our results show significant improvement in the indicators for bowel function after the 4 weeks of TPTNS. This significant improvement was also seen when comparing the pre-TPTNS period with the last assessment of the study, 4 weeks after the end of the intervention, demonstrating that the effects of TPTNS on intestinal function are sustained, even after it was discontinued. On the other hand, the absence of significant differences for the indicators of intestinal function when the data after 4 and 8 weeks of TPTNS were compared demonstrates that the effect of TPTNS does not seem to be cumulative, with no significant increase in the indicators when evaluated after more extended periods of intervention.

Bowel function was assessed by weekly frequency of defecation, stool consistency according to the mBSFS-C, and BF-S values. Stool consistency has a more significant correlation with intestinal transit time than frequency and can be determined simply and objectively through the application of graphic scales, such as the mBSFS-C [41]. This scale has been adapted to be applied directly to the pediatric population [27,28,29]. Most of the patients in the pre-intervention assessment presented type 2 stools, considered large and dry, and after the intervention, type 3 stools, soft and smooth. The BF-S is a questionnaire developed by Rintala and Lindahl [42] to evaluate the intestinal functional results of benign anorectal conditions. This instrument, consisting of 7 items, examines issues related to urgency, anorectal sensitivity, frequency of defecation, soiling, and social impact related to defecations. The maximum value of this score is 20. Healthy subjects have a mean score of 19.1 ± 1.3 points [30,42]. Most patients went from values close to 16 in the pre-intervention period to deals close to 18 or 19 in the post-TPTNS periods.

There was a significant improvement in the symptoms “effort to defecate” and “pain to defecate”, identified in the comparisons between the pre-TPTNS assessment and after 4 weeks of TPTNS, and between the pre-TPTNS assessment and 4 weeks after the end of the intervention, again indicating the potential for the benefits of TPTNS to be sustained, even after discontinuation. This improvement is not seen in the comparative analyses between the moments after 4 and after 8 weeks of TPTNS, reinforcing the hypothesis that the effect of TPTNS is not cumulative, according to the intervention time. There was no statistical improvement in the symptoms of “soiling” and “abdominal pain”. This can be explained by the limited number of patients who had fecal leaks before the beginning of the intervention, possibly because they were children who were already on medical treatment and who had previously undergone fecal disimpaction and were in regular use of laxatives. In addition, abdominal pain represents a less specific symptom, which can be influenced by other factors beyond those related to bowel habits.

The assessment of the quality of life is increasingly being discussed in the literature, with applicability in different clinical practice situations and scientific studies. It enables measuring health perception, as well as the impact of diseases, in the physical, psychological, and social aspects. This assessment is usually carried out using questionnaires. PedsQL 4.0 is an instrument for a generic assessment of the quality of life, which assesses several domains and can be used in healthy children and adolescents or many diseases, chronic or acute. It is a practical and brief instrument that children and guardians can answer [31,32]. In our study, we chose to apply it to the guardians since the children would fill out other questionnaires during the semi-structured interviews.

A long duration of symptoms and a low response to conventional treatment result in a significant impact on the child’s quality of life [2]. In our study, we noticed that the positive effects of TPTNS were accompanied by a significant improvement in the overall quality of life, both in total assessment and in the stratification by physical and psychosocial domains, as assessed by the PesdsQL 4.0 Questionnaire. These results were found on the same visits when significant improvements in indicators of bowel function and symptoms were identified, demonstrating that the effects of the intervention on quality of life can be sustained, even after stopping the intervention. On the other hand, once again, there were no significant differences when the moments after 4 and 8 weeks of TPTNS were compared, suggesting that the effect of TPTNS does not seem to be cumulative on overall quality of life either, with no significant increase when assessed over more extended periods of intervention.

The AQLCAFI Questionnaire was created to assess the quality of life related to fecal continence in patients with anorectal anomalies and Hirschsprung’s disease [33,34]. This questionnaire was validated for Brazilian children aged between 7 and 19 years. As the symptoms associated with constipation are similar to those presented by this group of patients, including the frequent possibility of fecal leaks due to overflow, we chose to use this instrument to assess the quality of life in our population of children with intractable intestinal constipation, since this was the only validated instrument available in Brazil to determine the quality of life associated with this type of clinical symptoms. In the assessment using this instrument, we found significant improvement in the final AQLCAFI indicators in all comparison scenarios, demonstrating a positive effect of TPTNS on quality of life-related explicitly to bowel habits. In the stratification by domains, a significant improvement was identified in the fields “lifestyle”, “behavior”, and “embarrassment” in the exact comparisons between the moments in which there were significant differences for the indicators of intestinal function, symptoms, and overall quality of life by the PedsQL 4.0, demonstrating that the effects of the intervention on the quality of life related to the intestinal function are also lasting, but not cumulative. According to the intervention time for the “depression” domain, it was found that; the positive effect of TPTNS took longer to be identified, seen only when comparing the moments 4 and 8 weeks post-TPTNS, and was sustained when comparing the moments pre-TPTNS and 4 weeks after the end of TPTNS, and 4 weeks post-TPTNS compared to 4 weeks after the end of the intervention, showing that this domain includes topics that take longer to reflect improvement in quality of life, consequent to improvement in bowel symptoms.

TPTNS proved to be an easily applicable treatment for daily use at home after properly training caregivers, who learned, without significant difficulties, to place the electrodes and turn on the device for the electrostimulation sessions. This was reflected in the high rate of patient compliance to the study, close to 80%, a value commonly considered adequate for compliance with pharmacological treatments [43]. The main difficulty pointed out by those responsible for carrying out the procedure was not related to the technique itself, nor the acceptance of the procedure by the patients, but to the availability of time to carry out the daily sessions. Most guardians classified the experience as good or excellent, and there were no adverse effects related to TPTNS during any study phase.

Some limitations of this study deserve to be highlighted. Although we followed the expected sample size, this is a single-center study with a limited number of patients and little diversity of demographic characteristics, which did not include patients from other age groups. These aspects may limit the generalizability of the findings to a broader population. Furthermore, the primary outcomes were assessed using validated but subjective instruments, including a stool consistency scale, bowel function score, and quality of life questionnaires. Objective measures, such as physiological markers, could strengthen study results. Other limitations include the lack of a control group, randomization, and comparisons with other existing treatments for functional constipation. These methodological limitations are justified by the difficulties in implementing a sham-control group in children with a daily procedure at home, supervised by their caregiver. A different methodological design, with daily sessions, randomized into intervention and sham-control groups, led by a research group member, would require the patients to travel daily to the hospital, running into financial and even ethical limitations. In addition, in our study, each patient can be considered a “control” of themselves since they were evaluated at different times, characterizing a paired sample. Supervised training sessions were scheduled to minimize the potential bias caused by caregivers’ application of the TPTNS, and the physiotherapist in charge was always available over the telephone to clarify any doubts and schedule an additional assessment as needed. Another limitation is related to the clinical scenario experienced by the patients during the study, with maintenance and possible changes in laxatives used. To minimize this potential bias, we monitored all modifications proposed by the medical team. We excluded from the results analysis the patients who eventually had an increased dose of laxatives from when the medical plan was changed. It should also be noted that although our study has demonstrated positive effects detected after four weeks of intervention and even four weeks after the end of the intervention, the optimal duration and frequency of TPTNS for sustained benefits were not determined and remain unclear. All these limitations must be considered in future research to contribute to a more comprehensive understanding of the effectiveness and broader implications of Transcutaneous Posterior Tibial Nerve Stimulation in treating functional constipation in the pediatric population.

On the other hand, this is the first study to analyze the effects and applicability of TPTNS in children with functional constipation for an adequate intervention and follow-up time. The patients were rigorously monitored and reassessed in at least three assessment moments. In addition, patients were assessed using different instruments, enabling sustained assessments of bowel function, including stool consistency, constipation-related symptoms, and overall quality of life-related to bowel symptoms. Another peculiar aspect is that the study included children with severe functional constipation, considered intractable, highly symptomatic, with poor response to conventional constipation treatment, and who had had symptoms for several years.

## 5. Conclusions

TPTNS proved to be an applicable and valuable method, which brought positive results through a significant improvement in defecation, intestinal symptoms, and health-related quality of life when applied as an adjuvant treatment in a sample of children with intractable constipation. The positive effects of this intervention are seen relatively early, detected after 4 weeks of intervention, and even 4 weeks after the end of the intervention. Therefore, this is an up-and-coming method that can be used as one of the available modalities for treating functional constipation in children and adolescents.

## Figures and Tables

**Figure 1 biomedicines-12-00164-f001:**
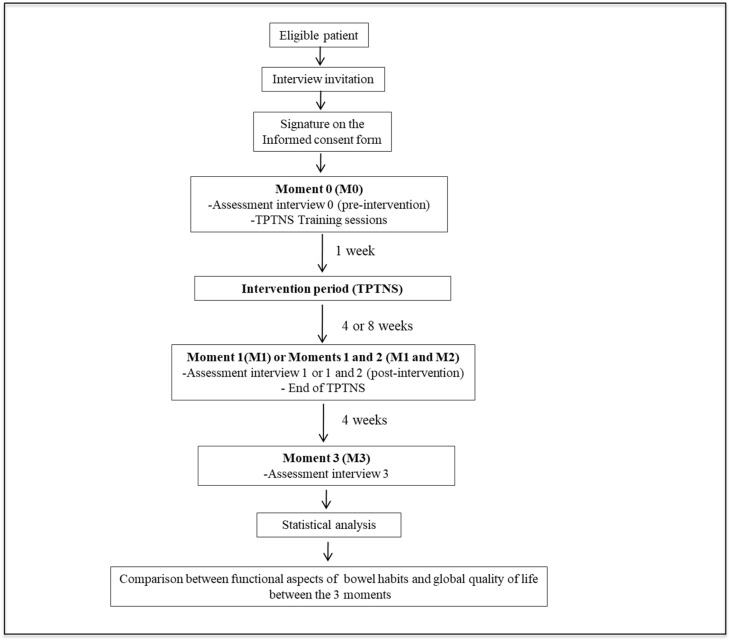
Flowchart of the study steps.

**Figure 2 biomedicines-12-00164-f002:**
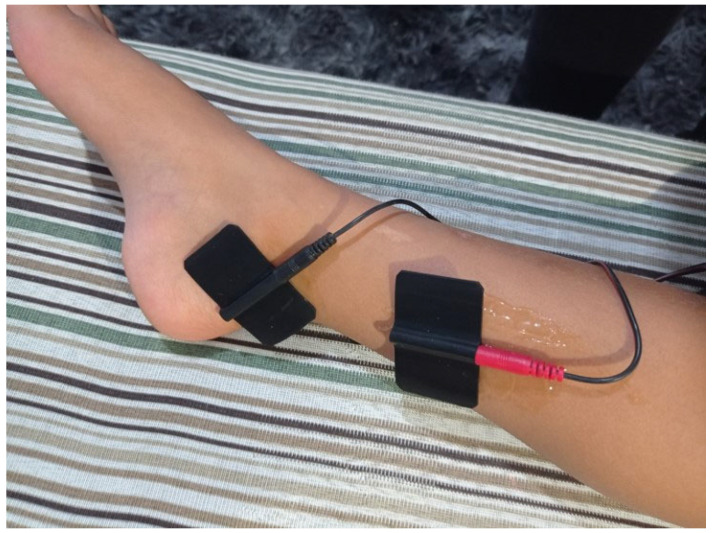
Electrode positioning in the patient’s leg.

**Figure 3 biomedicines-12-00164-f003:**
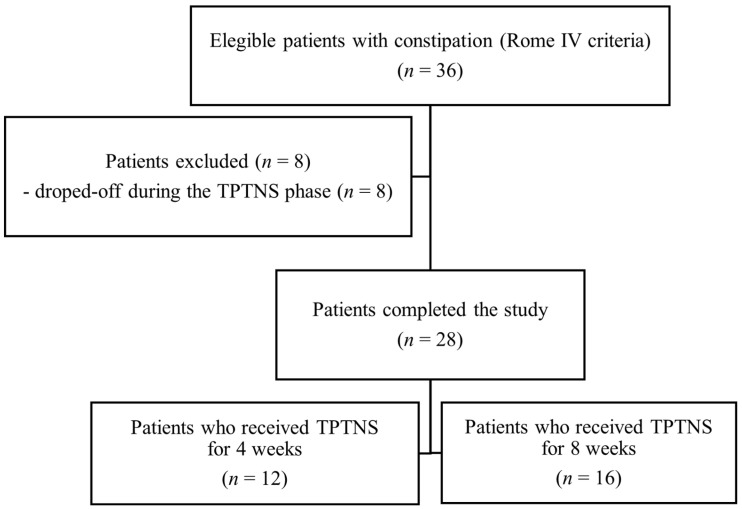
Flowchart of eligibility, inclusion, and exclusion criteria for the study patients.

**Table 1 biomedicines-12-00164-t001:** Sociodemographic characteristics of the patients who completed the study (*n* = 28).

Variable	
Children’s characteristics	
Gender	
Female	16 (57.2)
Male	12 (42.8)
Age (mo)	134 (105–145)
Firstborn child	15 (53.6)
Parents’ characteristics	
Maternal age (yr)	36 (32–40)
Paternal age (yr)	40 (33.5–45.5)
Maternal schooling (yr)	11 (9.5–11)
Paternal schooling (yr)	11 (8–11)
Number of children	2 (1–3)
Caregiver/Respondent	
Mother	22 (78.6)
Grandmother	4 (14.3)
Other	2 (7.1)
Housing characteristics	
Number of rooms	5 (5–6)
Number of people	4 (3–5)
Agglomeration index	1

The values are expressed as numbers (%) or medians (range).

**Table 2 biomedicines-12-00164-t002:** Clinical characteristics of the patients who completed the study (*n* = 28).

Variable	
Symptoms duration (mo)	84 (36–120)
Characteristics of defecation	
Hard bowel movements	23 (82.1)
Painful bowel movements	21 (75.0)
Stools characteristics	
mBSFS-C	2 (2–2)
Number of defecations per week	2 (1–3)
Bloody stools	8 (28.5)
Fecal incontinence	5 (17.9)
Associated symptoms	
Vomiting	4 (14.3)
Anorexia	5 (17.9)
Abdominal pain	12 (42.8)

The values are expressed as numbers (%) or medians (range). mBSFS-C: Modified Bristol Stool Form Scale for Children.

**Table 3 biomedicines-12-00164-t003:** Use of laxatives by the patients who completed the study (*n* = 28).

Variable	*n* (%)
Maintenance of the medication dose	17 (60.7)
Interruption of all laxatives	3 (10.7)
Decrease in laxative dose	4 (14.3)
Increase in laxative dose	4 (14.3)

**Table 4 biomedicines-12-00164-t004:** Comparison of bowel function and constipation symptoms in the different assessment moments.

Variable	M0 × M1	*p*-Value	M1 × M2	*p*-Value	M0 × M3	*p*-Value	M1 × M3	*p*-Value
mBSFS-C ^1^	2 (2/2) × 3 (2/3)	0.001	3 (2/3) × 3 (2/3)	0.680	2 (2/2) × 3 (2/3)	0.016	3 (2/3) × 3 (2/3)	0.506
Number of defecations/week ^1^	2 (2/3) × 6 (3.5/7)	0.0002	7 (4/7) × 7 (4/7)	1.000	2 (2/3) × 7 (3.25/7)	0.0009	7 (3.5/7) × 7 (3/7)	0.414
BF-S ^1^	16 (14/17) × 18 (17/19.5)	<0.0001	18 (17/20) × 19 (17/19.2)	0.313	16 (14.25/17) × 18 (17/19)	0.0001	18 (17/20) × 18 (17/19)	0.893
Hard bowel movements ^2^	81.4% × 22.2%	<0.001	18.7% × 12.5%	1.000	77.3% × 4.5%	<0.001	21.7% × 4.3%	0.125
Painful bowel movements ^2^	74.0% × 7.4%	<0.001	0 × 6.25%	1.000	68.2% × 9.1%	<0.001	4.3% × 8.7%	1.000
Incontinence ^2^	18.5% × 11.1%	0.625	12.5% × 6.2%	1.000	13.6% × 9.1%	1.000	8.7% × 8.7%	1.000
Abdominal pain ^2^	44.4% × 14.8%	0.021	12.5% × 12.5%	1.000	45.4% × 22.7%	0.179	17.4% × 26.0%	0.500

^1^ The values are expressed as medians (1st quartile/3rd quartile), *p*-value for the Wilcoxon test; ^2^ The values are expressed as percentages, *p*-value for the McNemar test. M0: Moment 0 (1 week before the beginning of the intervention); M1: Moment 1 (immediately after the period of 4 weeks of intervention); M2: Moment 2 (immediately after the period of 8 weeks of intervention, for patients who chose this duration of intervention); M3: Moment 3 (4 weeks after the end of the intervention period); mBSFS-C: Modified Bristol Stool Form Scale for Children; BF-S: Bowel Function Score.

**Table 5 biomedicines-12-00164-t005:** Comparison between the values of quality-of-life indicators assessed in the different assessment moments.

Variable	M0 × M1	*p*-Value	M1 × M2	*p*-Value	M0 × M3	*p*-Value	M1 × M3	*p*-Value
PedsQL 4.0 Physical	87.5 (81.2/98.4) × 93.7 (87.5/100) ^a^	0.029 ^c^	93.7 (81.2/100) × 96.8 (85/100) ^a^	0.635 ^c^	87.5 (81.2/96.8) × 96.8 (93.7/100) ^a^	<0.001 ^c^	93.7 (84.3/100) × 96.8 (93.7/100) ^a^	0.059 ^c^
PedsQL 4.0 Psychosocial	74.4 ± 12.9 × 81.8 ± 10.8 ^b^	<0.001 ^d^	85.8 (80/95) × 90 (83.3/93.3) ^a^	0.484 ^c^	75.3 ± 13.7 × 85.2 ± 10.2 ^b^	0.003 ^d^	83.3 (70/93.5) × 88.3 (77.7/92.5) ^a^	0.180 ^c^
PedsQL 4.0 Total	78.3 ± 12.0 × 85.0 ± 7.6 ^b^	<0.001 ^d^	86.4 (83.9/90.2) × 89.2 (86.3/92.9) ^a^	0.278 ^c^	79.3 (73/88.5) × 89.6 (79.3/94) ^a^	0.001 ^c^	85.8(79.3/90.2) × 90.2(80.9/94.5) ^a^	0.085 ^c^
AQLCAFI Lifestyle	3.7 (3.4/4) × 4 (3.7/4) ^a^	0.021 ^c^	3.9 (3.5/4) × 4 (3.8/4) ^a^	0.260 ^c^	3.7 (3.3/4) × 4 (3.8/4) ^a^	<0.001 ^c^	4 (3.7/4) × 4 (3.8/4) ^a^	0.239 ^c^
AQLCAFI Behavior	3.2 (3.1/3.5) × 3.5 (3.3/3.9) ^a^	<0.001 ^c^	3.5 (3.2/3.8) × 3.7 (3.5/4) ^a^	0.169 ^c^	3.2 (3.1/3.5) × 3.7 (3.6/4) ^a^	<0.001 ^c^	3.5 (3.3/3.8) × 3.7(3.5/4) ^a^	0.034 ^c^
AQLCAFI Depression	2.7 (2.5/3) × 2.8 (8.7/3) ^a^	0.394 ^c^	3 (2.6/3.1) × 3.5 (2.9/3.6) ^a^	0.001 ^c^	2.8 (2.6/3) × 3.4 (3.4/3.7) ^a^	<0.001 ^c^	2.8 (2.7/3.1) × 3.4 (3.4/3.7) ^a^	<0.001 ^c^
AQLCAFI Embarrassment	2.2 (2.2/3) × 3.3 (3/4) ^a^	<0.001 ^c^	3.3 (3/4) × 3.8 (3/4) ^a^	0.380 ^c^	2.3 (2.2/2.9) × 3.6 (3/4) ^a^	<0.001 ^c^	3.3 (3/4) × 3.6 (3/4) ^a^	0.687 ^c^
AQLCAFI Final	12.2 (11.5/13.2) × 13.4 (13/14) ^a^	<0.001 ^c^	13.1 (12.7/14.1) × 14.6 (14/15.1) ^a^	0.001 ^c^	12.4 ± 0.9 × 14.6 ± 0.7 ^b^	<0.001 ^d^	13.5 (13/14) × 14.9 (14.2/15.2) ^a^	<0.001 ^c^

^a^ The values are expressed as medians (1st quartile/3rd quartile); ^b^ The values are expressed as means ± standard deviation; ^c^
*p*-value for the Wilcoxon test; ^d^
*p*-value for the paired *t*-test. M0: Moment 0 (1 week before the beginning of the intervention); M1: Moment 1 (immediately after the period of 4 weeks of intervention); M2: Moment 2 (immediately after the period of 8 weeks of intervention, for patients who chose this duration of intervention); M3: Moment 3 (4 weeks after the end of the intervention period); PedsQL 4.0: Pediatric Quality of Life Inventory, version 4.0; AQLCAFI: Assessment of Quality of Life in Children and Adolescents with Fecal Incontinence.

**Table 6 biomedicines-12-00164-t006:** Results from the questionnaire to assess TPTNS applicability (*n* = 28).

Assessment of Applicability	*n* (%)	95% CI
Assessment of the experience		
Excellent	5 (17.9)	1.8–35.6
Good	18 (64.2)	45.2–79.3
Regular	5 (17.9)	1.8–35.6
Bad	0 (0)	0–1.2
Awful	0 (0)	0–1.2
Is the procedure considered difficult?		
Yes	2 (7.1)	1.9–22.6
No	26 (92.9)	77.3–98.2
The most important difficulty pointed		
Electrodes placing	0 (0)	0–1.2
Turning the device on and off	2 (7.1)	1.9–22.6
Procedure acceptance by the child	2 (7.1)	1.9–22.6
Finding time for it	7 (25.0)	12.7–43.3
Session duration	0 (0)	0–1.2
None	17 (60.7)	42.4–76.4
Was there pain during the procedure?		
Yes	0 (0)	0–1.2
No	28 (100)	87.9–100

TPTNS: transcutaneous posterior tibial nerve stimulation; CI: confidence interval.

## Data Availability

Data are contained within the article and Appendix A.

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
