# Peer review of "Transcutaneous Posterior Tibial Nerve Stimulation: An Adjuvant Treatment for Intractable Constipation in Children"

_biomedicines, 2024, doi:10.3390/biomedicines12010164_

Round 1
Reviewer 1 Report
Comments and Suggestions for Authors
Dear Sirs
This is an interesting study addressing the use of transcutaneous posterior tibial nerve stimulation: an adjuvant treatment for intractable constipation in children. The subject is relevant and up to date. The study is well designed, the methods are appropriated and, although not a prospective controlled trial, its conclusions about the safety and efficacy of the method are well based by their results.
It is believed that this manuscript is ready for publication
Author Response
Dear reviewer,
We want to thank you for your attention to our manuscript and for your pertinent comments and constructive criticisms.
Sincerely yours,
Reviewer 2 Report
Comments and Suggestions for Authors
Manuscript details:
Journal: Biomedicines
Manuscript ID: biomedicines-2792028
Title: Transcutaneous posterior tibial nerve stimulation: an adjuvant
treatment for intractable constipation in children
Authors: Rebeca Mayara Padilha Rego, Nilton Carlos Machado, Mary de Assis
Carvalho, Johann Souza Graffunder, Crhistiano Fraguas, Erika Veruska Paiva
Ortolan, Pedro Luiz Toledo de Arruda Lourenção *
Present work the authors aim to evaluate the clinical results and applicability of TPTNS as an
adjuvant treatment for children and adolescents with functional constipation. The authors found that TPTNS is an adjuvant treatment for intractable functional constipation, improving bowel function and quality of life. The effects of TPTNS were observed relatively early and sustained even after treatment cessation. Even though this manuscript is interesting, a some general concern needs to be addressed before further consideration.
Overall comment:
-The study involved a relatively small sample size of 38 patients, which may limit the generalizability of the findings to a broader population.
-The research was conducted in a single-center, which could introduce bias and reduce the external validity of the results. Multi-center studies are needed to enhance the robustness of the findings.
-The study utilized an uncontrolled, prospective cohort design, lacking a control group for comparison. This design choice makes it challenging to establish a direct causal relationship between Transcutaneous Posterior Tibial Nerve Stimulation (TPTNS) and the observed improvements.
-The follow-up period after the intervention was relatively short, lasting only 4 weeks. A more extended follow-up duration could provide insights into the long-term efficacy and sustainability of the observed benefits.
-The study primarily relied on subjective measures such as patient-reported improvements in stool consistency, frequency of defecation, and quality of life. Objective measures, such as physiological markers, could strengthen the study's outcomes.
- The study may lack diversity in terms of patient demographics, potentially limiting the applicability of the results to a more diverse population.
- Adherence to TPTNS was reported to be approximately 78%, which could be subject to reporting bias. Objective measures of adherence and potential influencing factors were not extensively explored.
- While the study demonstrated the positive effects of TPTNS, it did not compare the intervention with other existing treatments for functional constipation, limiting the ability to assess its relative effectiveness.
- Although the study demonstrated positive effects after 4 weeks of TPTNS, the optimal duration and frequency of TPTNS for sustained benefits remain unclear.
- The study focused on children and adolescents, and the applicability of TPTNS to other age groups remains uncertain.
Therefore, addressing these limitations in future research will contribute to a more comprehensive understanding of the effectiveness and broader implications of Transcutaneous Posterior Tibial Nerve Stimulation in the treatment of functional constipation in the pediatric population.
Comments on the Quality of English LanguageMinor editing of English language required
Author Response
Dear reviewer,
We want to thank you for the pertinent comments and constructive criticisms.
We have included the answers to your comments, point by point, below.
Modifications made according to the reviewers' suggestions are highlighted in red in the revised version of the manuscript.
Thank you very much in advance for your kind attention,
Sincerely yours,
# 1) The study involved a relatively small sample size of 38 patients, which may limit the generalizability of the findings to a broader population.
Although we followed the expected sample size, we agree that our sample is small, which may limit the generalizability of the findings to a broader population. This study limitation was added in the Discussion section on page 12.
# 2) The research was conducted in a single-center, which could introduce bias and reduce the external validity of the results. Multi-center studies are needed to enhance the robustness of the findings.
We agree with this study limitation. We also add this comment in the Discussion section on page 12.
# 3) The study utilized an uncontrolled, prospective cohort design, lacking a control group for comparison. This design choice makes it challenging to establish a direct causal relationship between Transcutaneous Posterior Tibial Nerve Stimulation (TPTNS) and the observed improvements.
We also agree with these limitations, also presented in the Discussion section on page 12. These methodological limitations occurred due to the difficulties in implementing a sham-control group in children with a daily procedure at home, supervised by their caregiver. A different methodological design, with daily sessions, randomized into intervention and sham-control groups, led by a research group member, would require the patients to travel daily to the hospital, running into financial and ethical limitations. In addition, in our study design, each patient can be considered a "control" of themselves since they were evaluated at different times, characterizing a paired sample.
# 4) The follow-up period after the intervention was relatively short, lasting only 4 weeks. A more extended follow-up duration could provide insights into the long-term efficacy and sustainability of the observed benefits.
We agree that the follow-up period after the intervention was relatively short, lasting only four weeks, and that a more extended follow-up could provide insights into the long-term efficacy and sustainability of the observed benefits. However, this is the first study that evaluated TPTNS for children with constipation, including an intervention time higher than ten days and a reassessment period four weeks after the end of the intervention.
# 5) The study primarily relied on subjective measures such as patient-reported improvements in stool consistency, frequency of defecation, and quality of life. Objective measures, such as physiological markers, could strengthen the study's outcomes.
Although we used validated instruments, we agree they are subjective and that objective measures, such as physiological markers, could strengthen study results. We also add this comment in the Discussion section on page 12.
# 6) The study may lack diversity in terms of patient demographics, potentially limiting the applicability of the results to a more diverse population.
We agree with this study limitation. We also add this comment in the Discussion section on page 12.
# 7) Adherence to TPTNS was reported to be approximately 78%, which could be subject to reporting bias. Objective measures of adherence and potential influencing factors were not extensively explored.
We agree that our study did not extensively explore objective adherence measures and potential influencing factors. This is related to the study design, which includes daily procedures performed at home and supervised by caregivers. To minimize this potential bias, we conducted regular weekly supervised training sessions, and the physiotherapist in charge was always available over the telephone to clarify any doubts and schedule an additional assessment as needed. These limitations were also presented in the Discussion section on page 12.
# 8) While the study demonstrated the positive effects of TPTNS, it did not compare the intervention with other existing treatments for functional constipation, limiting the ability to assess its relative effectiveness.
We agree with this study design limitation. We also add this comment in the Discussion section on page 12.
# 9) Although the study demonstrated positive effects after 4 weeks of TPTNS, the optimal duration and frequency of TPTNS for sustained benefits remain unclear.
We agree that although our study has demonstrated positive effects detected after four weeks of intervention and even four weeks after the end of the intervention, the optimal duration and frequency of TPTNS for sustained benefits were not determined and remain unclear. We also add this comment in the Discussion section on page 12.
# 10) The study focused on children and adolescents, and the applicability of TPTNS to other age groups remains uncertain.
We agree with this study design limitation. We also add this comment in the Discussion section on page 12.
# 11) Therefore, addressing these limitations in future research will contribute to a more comprehensive understanding of the effectiveness and broader implications of Transcutaneous Posterior Tibial Nerve Stimulation in the treatment of functional constipation in the pediatric population.
We also agree with this critical comment. We added this note in the Discussion section on pages 12 and 13.
Reviewer 3 Report
Comments and Suggestions for Authors
Well written, interesting paper regards TPTNS on what can be a very difficult problem
A photo of the device and the leg attachments would be great
The methods say the therapy was 30 minutes each day; why did 10 patients not finish the study (38-28=10) ? The methods start with 38, then the results say 36 started. Is that 38 were invited, 36 started?
Please explain how 38 got to 28 in more detail please?
Were there any side-effects of the therapy?
Did the electrodes stay adhered?
Did any patients complain of pain?
Author Response
Dear reviewer,
We want to thank you for the pertinent comments and constructive criticisms. We have included the answers to your comments, point by point, below.
Modifications made according to the reviewers' suggestions are highlighted in red in the revised version of the manuscript.
Thank you very much in advance for your kind attention,
Sincerely yours,
# 1) A photo of the device and the leg attachments would be great
In Figure 2, we added a photograph of the electrodes in the patient's leg.
# 2) The methods say the therapy was 30 minutes each day; why did 10 patients not finish the study (38-28=10)? The methods start with 38, then the results say 36 started. Is that 38 were invited, 36 started?
Please explain how 38 got to 28 in more detail please?
We apologize for the error regarding the number of patients included in the study. Thirty-eight patients were eligible and were invited to participate, but 36 agreed to participate in the study. Eight patients were dropped off during the TPTNS phase. The caregivers of these eight patients explained that they did not have time to supervise the intervention and stopped carrying it out. No patient had side effects or pain during the TPTNS.
Thus, of the 36 patients who started the study, eight dropped out during the intervention phase, and 28 completed the study (12 patients who received TPTNS for four weeks and 16 patients who received TPTNS for eight weeks).
We corrected the number from 38 to 36 patients in the Abstract, Methods, and Results sections. We added the information regarding the patients who dropped out of the study in the Results section (item 3.3. Analysis of TPTNS applicability) on page 10.
# 3) Were there any side-effects of the therapy?
During the study, no adverse effects of the intervention were reported or identified. This information was added in the Results section on page 10.
# 4) Did the electrodes stay adhered?
The electrodes stayed adhered during the posterior tibial nerve stimulation, simultaneously on both lower limbs, for 30 minutes daily. After each TPTNS session, the electrodes were removed. The self-adhesive silicone electrodes were changed every ten sessions to minimize the potential adverse effects of the therapy. This information was detailed in the method section on page 4.
# 5) Did any patients complain of pain?
No patient complained of pain during the sessions. This information was added in the Results section on page 10.
Round 2
Reviewer 2 Report
Comments and Suggestions for Authors
The authors are grateful for the incorporation of the suggested changes into the paper, which is now ready for publication.
Congratulations!!!